# A Family with Myh7 Mutation and Different Forms of Cardiomyopathies

**DOI:** 10.3390/biomedicines11072065

**Published:** 2023-07-22

**Authors:** Bianca Iulia Catrina, Florina Batar, Georgiana Baltat, Cornel Ioan Bitea, Andreea Puia, Oana Stoia, Sorin Radu Fleacă, Minodora Teodoru

**Affiliations:** 1Department Basic Science-Physiopathology, Faculty of Medicine, “Lucian Blaga” University, 550169 Sibiu, Romania; catrina.bianca@ymail.com; 2County Clinical Emergency Hospital of Sibiu, 550245 Sibiu, Romania; cornelioanbitea@yahoo.com (C.I.B.); puiaandreea@yahoo.com (A.P.); oana.stoia@ulbsibiu.ro (O.S.); radu.fleaca@ulbsibiu.ro (S.R.F.); minodora.teodoru@ulbsibiu.ro (M.T.); 3Department Basic Science-Physiology, Faculty of Medicine, “Lucian Blaga” University, 550169 Sibiu, Romania; 4Department Clinic-Medicine, Faculty of Medicine, “Lucian Blaga” University, 550196 Sibiu, Romania; baltat.georgi@gmail.com; 5Department Medicine-Internal Medicine, Faculty of Medicine, “Lucian Blaga” University, 550169 Sibiu, Romania; 6Department of Surgery, Faculty of Medicine, “Lucian Blaga” University, 550169 Sibiu, Romania

**Keywords:** hypertrophic cardiomyopathy, MHY7 gene, implantable cardioverter defibrillators

## Abstract

Background: Hypertrophic cardiomyopathy (HCM) and dilated cardiomyopathy (DCM) are common heart muscle disorders that are caused by pathogenic variants in sarcomere protein genes. In this study, we describe a variant in the MHY7 gene, segregating in a family having three different phenotypes of cardiomyopathies. MYH7 encodes for the myosin heavy-chain β (MHC-β) isoform involved in cardiac muscle contractility. Method and results: We present the case of a family with four members diagnosed with HCM and four members with DCM. The proband is a 42-year-old man diagnosed with HCM. He has an extended family of eight siblings; two of them are diagnosed with HCM and are implantable cardioverter–defibrillator (ICD) carriers. One of the siblings died at the age of 23 after suffering a sudden cardiac arrest and DCM of unknown etiology which was diagnosed at autopsy. Another brother was diagnosed with DCM during a routine echocardiographic exam. Genetic testing was performed for the proband and two of his siblings and a niece of the proband, who suffered a cardiac arrest at the age of nine, all being MYH7 mutation positive. For all four of them, cardiac imaging was performed with different findings. They are ICD carriers as well. Conclusions: Our results reveal three variants in phenotypes of cardiomyopathies in a family with MYH7 mutation associated with high SCD risk and ICD needed for primary and secondary prevention.

## 1. Introduction

Hypertrophic cardiomyopathy (HCM) and dilated cardiomyopathy (DCM) are congenital heart muscle disorders. HCM is characterized by cardiac hypertrophy (increased chamber wall thickness) that is associated with increased ventricular contractility and impaired relaxation. DCM is defined as enlarged ventricular volume with contractile impairment [1]. In contrast to HCM, DCM is characterized by a thinner-than-normal ventricular wall and can culminate in heart failure. Left ventricular non-compaction cardiomyopathy (LVNC) is a type of cardiomyopathy that is characterized by excessive trabeculations of the left ventricle (LV) [2].

In many of the families afflicted with either disease, responsible mutations have been identified in various sarcomeric proteins [3]. Pathogenic proteins linked to HCM serve a variety of purposes, including those related to enzymatic activity and force generation (β-cardiac myosin, myosin light chains, actin), structural scaffolding (myosin-binding protein C, actin, titin, desmin), and regulatory activities (tropomyosin, troponin, and myosin-binding protein C. Although the genetics of DCM mutations are more varied, mutations in β-cardiac myosin are included in the list of genes that cause this disorder, along with other mutations in the sarcomeric protein genes [4].

The gene encoding the beta-myosin heavy chain (MYH7) is known to make a protein that is found in cardiac muscle and skeletal muscle fibers [5]. Approximately 200 mutations of this gene are known to cause DCM and HCM. Out of these, around 30 mutations are responsible for causing left ventricular noncompaction. Although efforts have been made to link genotype and phenotype in order to support clinically oriented diagnostic and risk factor assessment algorithms, the data sets on individual mutations are still insufficient to serve as substantial clinical guidance. Arg403Gln, Arg453Cys, and Arg719Trp are three specific mutations in MYH7 that appear convincingly associated with adverse outcomes; however, data suggest that at-risk patients carrying these mutations also exhibit clinical risk factors at the time of events, limiting the additional prognostic benefit of genetic diagnosis [6].

Additionally, since gene-based diagnosis is independent of clinical symptoms, it enables the accurate identification of at-risk individuals even before a diagnosis, in the preclinical stage. Before a clinical diagnosis can be made, relatives with a missense variant in the family might undergo genetic testing to see if they have inherited it [7].

## 2. Materials and Methods

The patients were admitted to the Academic County Hospital Sibiu between 2019 and 2021. Family members underwent clinical examination, which included general examination, electrocardiography using 24 h Holter monitoring, electrocardiogram (HM-ECG), Doppler echocardiography (ECHO), and cardiac magnetic resonance (cMRI). ECHO and cMRI imaging criteria for LVNC were applied, as previously suggested by Jenni and investigators [8]. Our study was conducted under the Declaration of Helsinki in its current form.

The echocardiographic studies were performed on a Philips Epic 7C ultrasound machine. M-mode or 2D imaging was used to assess the LV end-diastolic diameter (LVEDD) and LV end-systolic diameter (LVESD). The ejection fraction of the left ventricle (EFLV) was calculated using the modified Simpson’s biplane method. Systolic anterior movement (SAM) of the mitral valve was defined as absent (no contact with the septum) or present (contact between leaflet and septum during systole). The left ventricular outflow tract (LVOT) gradients were measured pre and immediately post a symptom-limited exercise test and patients were classified into non-obstructive (<30 mmHg at rest and exercise) and obstructive (≥30 mmHg at rest or exercise) groups [9]. The average of the peak longitudinal strain from a 16 LV segment model was used to define the term LV global longitudinal strain (GLS). The standard deviation of the duration to peak negative strain in 16 LV segments was used to quantify mechanical dispersion [10].

The use of strain analysis is recommended in HCM as a clinical tool for the evaluation of the LV systolic function. It has been shown that because of myocardial disarray and fibrosis, systolic contraction in HCM is heterogeneous, and mechanical dispersion assessed by two-dimensional strain reflects heterogeneous myocardial contraction [11].

Cardiac magnetic resonance (cMR) was performed on a 3-Tesla system (Ingenia 3.0 T Philips, Eindhoven, Holland) with contrast, gadobutrol 0.1 mL/kg (Gadovist, Bayer AG, Berlin, Germany).

ECHO and cMRI imaging criteria of LVNC were applied, as previously suggested by Jenni et al., including a bilayered myocardium, a noncompacted to compacted ratio > 2:1, communication with the intertrabecular space demonstrated using color Doppler, the absence of coexisting cardiac abnormalities, and the presence of multiple prominent trabeculations in end-systole [8].

HCM risk-sudden cardiac death (SCD) uses predictor variables that have been associated with an increased risk of sudden death in at least one published multivariable analysis. The model provides individualized 5-year risk estimates of major clinical features associated with an increased risk of sudden cardiac death in adults [12].

## 3. Results

Case report:

We present the case of a family with four members diagnosed with HCM and four members with DCM. The proband is a 42-year-old man diagnosed with HCM. He has an extended family of eight siblings; two of them are diagnosed with HCM and are implantable cardioverter–defibrillator (ICD) carriers. One of the siblings died at the age of 23 after suffering a sudden cardiac arrest, where DCM of unknown etiology was diagnosed at autopsy. Another brother was diagnosed with DCM during a routine echocardiographic exam. The last three brothers had negative echocardiographic and electrocardiographic screenings for both HCM and DCM. The 11-year-old proband niece was diagnosed with HCM and had an ICD implant after an SCD episode for secondary prevention. The proband’s father and uncle, diagnosed with DCM, died at 56 and 57 years old.

Genetic testing was performed for the proband and two of his siblings previously diagnosed with HCM, and for the proband’s niece; all four of them were MYH7 mutation positive.

The family’s four-generation pedigree is shown in Figure 1.

Next, we will present the cases of these three siblings that were examined in our center and confirmed with MYH7 mutation.

In Case 1, the proband (Figure 2E–G) is a 42-year-old male, without significant medical history, who was admitted for heart failure symptoms; imaging evaluation by transthoracic echocardiography (TTE) showed a dilated left ventricle of 67 mm, a mildly reduced ejection fraction, aspect of non-obstructive HCM (SAM absent), and moderate mitral regurgitation. Global longitudinal strain bull’s eye imaging showed reduced global longitudinal strain (−12.2%) with an ejection fraction of 41%, and 64 ms mechanical dispersion. Cardiac magnetic resonance (cMR) revealed aspects of non-obstructive HCM and criteria for left-ventricular noncompaction (with a 3.7 non-compacted/compacted ratio by Jenny criteria evaluated using CMR) [8]. The cardiac magnetic resonance imaging SSFP-cine sequence four-chamber view displayed bi-ventricular hypertrophy with asymmetric LV hypertrophy, IVS predominance (IVS 18 mm), an RV free wall of 9 mm, dilated left ventricle, and global hypokinesia. It was also noted that there was important trabeculation of the left ventricle apex, lateral, and anterior walls. Cardiac magnetic resonance imaging late gadolinium enhancement showed mid-wall enhancement in the IVS, LV apex, and RV (right ventricle) wall. 24 h Holter-EKG monitoring revealed polymorphic ventricular ectopies with high arrhythmic loading, and the 5-year-risk of SCD calculated with the 2014 ESC HCM-risk-SCD calculator was 5.48%. The patient had an ICD implant as primary prophylaxis for SCD. At the one-year follow-up, the ICD recordings showed four non-sustained VTs with no need for therapies; also, the patient suffered a minor ischemic stroke with no clear origin of the thrombus.

Case 2 (Figure 2A,B) is a 55-year-old woman presenting heart failure symptoms two years prior to the cardiovascular examination. The ECHO parasternal long-axis view revealed asymmetric left ventricle hypertrophy with the predominant interventricular septum (IVS) hypertrophy of a maximum of 22 mm. The resting LVOT gradient was 20 mmHg and the provoked gradient was 33 mmHg. The mitral valve had moderate regurgitation, with a vena contracta of 6.2 mm, and SAM was absent. Global longitudinal strain bull’s eye imaging showed normal global longitudinal strain (GLS) and preserved ejection fraction, with reduced regional longitudinal strain predominantly in the basal segments of the IVS and inferior wall. Despite a normal GLS value (−20.4%) and low values of mechanical dispersion (36 ms), the patient presented three episodes of VT. She was diagnosed with obstructive HCM and a 5-year risk of SCD of 4.02% calculated with the 2014 ESC HCM-risk-SCD score, and she received an implantable cardioverter–defibrillator (ICD) as primary prophylaxis. At the first year of follow-up, the device recordings showed three asymptomatic episodes of non-sustained VT, one of them treated with anti-pacing therapy (ATP). The important arrhythmic burden could be explained by the basal inferior segment of the LV that has a decreased regional GLS, probably because of important fibrosis. The ECHO should be correlated with cRM for better comprehension. Her 11-year-old daughter, diagnosed with HCM, had an ICD implanted when she was 9 years old for sustained polymorphic ventricular tachycardia (VT).

Case 3 (Figure 2C,D) is the oldest brother, a 47-year-old male with a history of surgical myomectomy (2014) for obstructive HCM and cryoablation for paroxysmal atrial fibrillation (2018). ECHO after the surgical myomectomy revealed in the parasternal long-axis view asymmetric left ventricle hypertrophy with predominant IVS hypertrophy of a maximum of 22 mm, SAM was absent, and mild eccentric mitral regurgitation. Global longitudinal strain bull’s eye imaging showed reduced GLS (−11.4%) despite preserved ejection fraction; the regional longitudinal strain was severely reduced in the mid and basal segments of the interventricular septum, left ventricle anterior, and inferior walls. Despite a low GLS value and a high value of mechanical dispersion of 80, the patient did not present episodes of VT. The low values of GLS and high values of mechanical dispersion could be explained by the myocardial scar secondary to myomectomy. The 5-year risk of SCD of 7.5% was calculated with the 2014 ESC HCM-risk-SCD calculator SCD and prophylaxis with an ICD implant was performed (2017). The ICD did not record any rhythm disturbance after ablation.

## 4. Discussions

Myosin heavy chain (MHC-β) isoform MYH7 is encoded by the MYH7 gene and is largely expressed in type 1 skeletal muscle fibers and the cardiac ventricle. This isoform is different from MYH6, the fast cardiac myosin heavy-chain isoform. The main protein that makes up the thick filament in cardiac muscle, MHC-β, is crucial to the contraction of this muscle. It is a crucial part of the sarcomere and performs through intricate protein interactions with other myosin and actin fiber tracks (thin filaments). Mutations (more than 200) in the MYH7 gene are usually associated with several structural cardiomyopathies, such as hypertrophic, restrictive (RCM), dilated (DCM) or left ventricular non-compaction (LVNC) as well as myosin storage myopathy [13].

Our results report the presence of an MYH7 gene mutation associated with the HCM, LVNC, and DCM phenotypes in one family. The clinical ramifications of these findings suggest that structural abnormalities are more common in MYH7 mutations. As these patients are more likely to develop mitral valve dysfunction and LVOT obstruction, they would likely benefit from more intensive imaging surveillance, which should begin at a young age. It is logical to assume that individuals with the MYH7 gene mutation would benefit from earlier initiation and more aggressive medical therapy regarding diastolic dysfunction [14].

In HCM, a heart condition linked to arrhythmias and sudden death, there are still unsolved questions regarding risk assessment and the efficacy of implanted cardioverter–defibrillator treatment. In HCM patients, myocardial fibrosis may contribute to arrhythmogenesis. The substrate for Vas (ventricular arrhythmias) is related to increased electrical dispersion and inhomogeneity of intraventricular conduction in HCM patients with malignant arrhythmias. This is likely due to variations in myocyte diameter, disorder, and myocardial fibrosis [15].

Cardiac conduction disease and ventricular arrhythmia rates are higher in HCM patients with MYH7 mutations in comparison to MYBPC3 [16]; MYH7 mutation leads to a more severe phenotype of HCM compared with patients without such a mutation. In secondary prevention, things are quite clear, and the indication of ICD is strongly recommended, but in primary prevention, the criteria need further refinement. The current risk score for HCM uses non-sustained ventricular tachycardia (nsVT), determined by Holter monitoring, as an important indicator of life-threatening conditions [12].

Echocardiography is central to the diagnosis and monitoring of HCM and is an excellent tool for assessing regional and global left-ventricular (LV) function, both systolic and diastolic, and determining associated abnormalities of the mitral valve and left-ventricular outflow tract. Halland and co. explored the relationship between mechanical dispersion and myocardial fibrosis in HCM and demonstrated that imaging evaluation (TTE and CRM) can be used in arrhythmic risk stratification [10]. In 150 HCM patients, LGE-CMR identified fibrosis, wall thickness, GLS, and mechanical dispersion as indicators of VAs. Mechanical dispersion is a significant and reliable predictor of arrhythmic events and is correlated with the degree of fibrosis. Despite a preserved EF, GLS is related to fibrosis and arrhythmic burden. Identification of HCM patients at high risk for VAs and SCD may be identified using mechanical dispersion and GLS.

Imaging evaluation has an important role in diagnosis, arrhythmic risk assessment, and family screening. CMR imaging is well-suited for identifying many phenotypic presentations of HCM, giving a diagnosis, risk prediction, and preprocedural planning for septal reduction. It also allows for the evaluation of myocardial fibrosis following contrast injection using LGE, which is used as a noninvasive marker for a high risk of ventricular tachyarrhythmias and HF development with systolic dysfunction (i.e., myocardial fibrosis) [9].

Reduced GLS, increased mechanical dispersion, and the presence of late gadolinium enhancement in cardiac magnetic resonance imaging have shown a correlation with arrhythmic risk, but there is still a need for further studies to validate cut-off values [10]. For these three cases, decision-making for receiving ICDs was based on the reduced ejection fraction, previous arrhythmic events with a high risk for sustained ventricular tachycardia or fibrillation, and the family history of sudden cardiac deaths at a young age.

The peculiarity of this family is the aggregation of three types of cardiomyopathies: HCM, DCM, and LVNC associated with high SCD risk: three of four DCM patients presented SCD and four of four HCM patients have ICD (three of four in primary prophylaxis and one in secondary prophylaxis). Of note is that the only patient with secondary prophylaxis is the youngest member of the family who tested positive for the MYH7 mutation.

Another particular aspect is the overlap of two phenotypes, HCM and LVNC, in the same patient, thus determining a combined risk for both arrhythmic and thromboembolic events. At the first-year follow-up, it was noted that the patient with the lowest score had the highest arrhythmic burden, while the patient with the highest score did not have any ventricular arrhythmia. Our opinion is that the current score is not infallible and that the workup for HCM patients should not resume at ECHO. Due to appropriate risk stratification and primary prevention using ICDs, sudden cardiac death decreased in patients with known HCM.

The disease progresses through a decline in left ventricular ejection fraction (<50%), termed “burned-out HCM” and heart failure symptoms [17].

## 5. Conclusions

Individuals with MYH7 mutations are prone to SCD and have a family history of cardiomyopathies, they also have more severe structural conditions. As shown in this family history, more types of cardiomyopathies can coexist in the same family even overlapping in the same patient. Early family screening once a member is diagnosed and ICD implantation as needed are stepping stones in the management of this pathology.

## Figures and Tables

**Figure 1 biomedicines-11-02065-f001:**
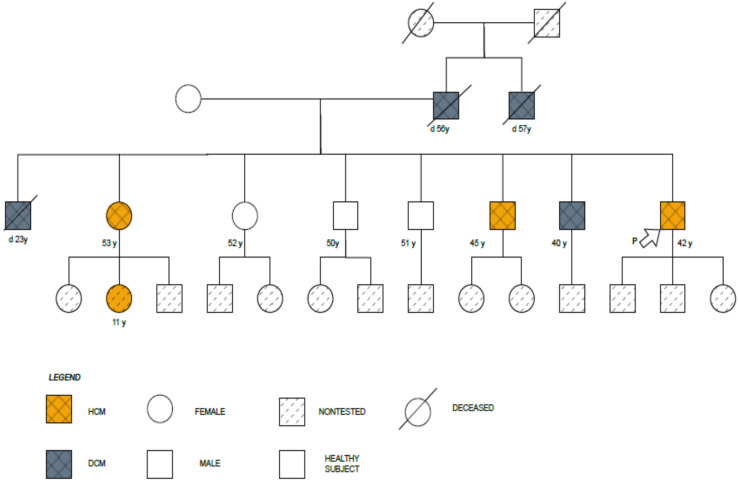
Family four-generation pedigree showing the presence of HCM or DCM.

**Figure 2 biomedicines-11-02065-f002:**
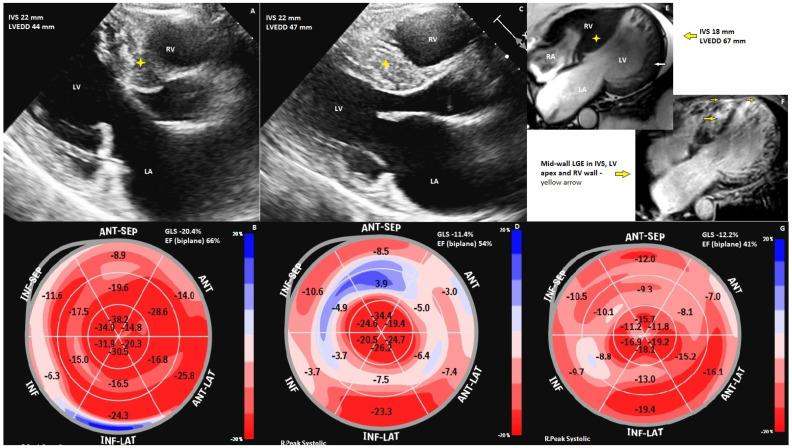
Imaging of hypertrophic cardiomyopathy of the three siblings. (**A**) ECHO of the 2nd patient, parasternal long axis 5 chambers; (**B**) ECHO of the 2nd patient, bull’s eye displays of GLS; (**C**) ECHO of the 3rd patient, parasternal long-axis view; (**D**) ECHO of the 3rd patient, bull’s eye displays of GLS; (**E**) cMR of the proband, showing the trabeculations in end-systole; (**F**) cMR of the proband, showing late gadolinium enhancement in LV wall, RV wall and interventricular septum; (**G**) ECHO of the proband, bull’s eye displays of GLS.

## Data Availability

Data is unavailable due to privacy and ethical restrictions.

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
