# Peer review of "A Family with Myh7 Mutation and Different Forms of Cardiomyopathies"

_biomedicines, 2023, doi:10.3390/biomedicines11072065_

Round 1
Reviewer 1 Report
This is a case report of a family with HCM and DCM and an (unknown) variant in the MYH7 gene. The manuscript has to be extensively rewritten as it cannot be accepted in the actual form. The abstract has to be better structured as it repeats family information and is very difficult to follow. This is a report on a genetic mutation in MYH7, HOWEVER, nowhere in the manuscript is there genetic information. What variant does the family carry? What is its classification? How were the genetic studies performed? The pedigree figure does not conform to standard rules and is impossible to understand. Some more comments in attached pdf.

Author Response
Dear reviewer,
Thank you for your time and your suggestion for improving our manuscript. We tried to reply to your requests. the extended information about the genetic information we could not fulfill because the tests were not done in our hospital, even in our country and we only have available the discharge letters from the hospital in Paris where the tasting was done. In those letters, it only confirms the MYH 7 mutation. If you need them I can send you the attachments.
We hope the manuscript is in better shape now.

Reviewer 2 Report
The paper is interesting, but its final form should be greatly improved.
Material and methods are not so clear and should be better explained. Quotation marks and english form should be checked. Echocardiographic and MR methods and machines should be reported in more detail and references should be also reported for derived measurements for each methodology.
Abbreviations should be explained throughout the paper when first reported (as example ECHO in page 2, Vas and VAs in page 6...).
It is not clear why to put a sub-paragraph number when there is only one paragraph: 2. Materials and Methods 2.1. Clinical Investigation of the Patients.
Figure 1 is not clear and should be better explained in the legend. If numbers are referring to age, why in the text the proband is reported as having 42 years, but 36 in the Figure?
Figure 2 is also not clear. Quadrants A,B,C,D etc. should be indicated and their total number should correspond to the text. In fact in the Figure only six quadrants are present, while letters indicated in the text are 7 (A,B,C,D,E,F,G) ?
The legend of Figure 2 should report more details and explain better what is represented in each quadrant. It should be also very clear which case is which in Figure 2 (proband? siblings? ). Reported Cases 1, 2 and 3 should be also indicated and identified better in Figure 1 and Figure 2, to better understand family genetic relation.
In Case 1 there are 3 repetitions of the same values for EF (41%) and GLS (-12.2%) and mechanical dispersion (64 ms). Please check and solve.
In Case 2: .. "GLS value (-20.4%) and the low values of mechanical dispersion (36 ms), the patient presented three episodes of VT". Its should be then discussed which could be the real clinical value of a normal GLS, when all the Literature is suggesting that and abnormal GLS should be indicating early subtle LV abnormalities. So in this situation probably GLS could be considered of no or minimal clinical value? Please indicate also some references for the discussion.
In the Discussion all the mutation should be better explained: .." ; MIM 192600), Restrictive (RCM), Dilated (DCM; MIM 115200), or Left Ventricular Non-Compaction (LVNC; MIM 613426), as well as myosin storage myopathy (MIM 608358). Moreover, deletions or missense mutations exclusively located within exons 32 to 36 cause Laing distal myopathy (MIM 160500) [11]. "
In the discussion the Authors reported: " Patients with MYH7 mutation would probably benefit from more intense imaging surveillance that should start at a younger age as they are likely to develop mitral valve dysfunction and LVOT obstruction. " . No apparent discussion of mitral valve dysfunction is present in the paper concerning the reported family.
Some references are reported after the final dot of the phrase.
References should be updated.
The risk prediction capability of the calculated score in each reported subject shuld be discussed. How long was the real life follow-up of these patients and which was the observed real life arrhythmic burden during the observed follow-up ? The difference between calculated risk score and real life arrhythmic burden could need some discussion.
Before the Conclusions the following sentence does not appear necessarily true: ..." the route to mortality commonly proceeds through the decline in left ventricular ejection fraction (LVEF) <50% ..."
Final sentence before Conclusions is also not clear: "Regardless of whether AF is the cause or effect of Burned-out HCM, it is strongly associated with symptoms, and declining LVEF or mitral regurgitation may prevent burnout by reducing wall stress."
Some sentences are confused and not clear. Check typing. As an example some references are written after the terminal dot of the phrase.
Author Response
Dear reviewer,
Thank you for your suggestion for improving our manuscript. We tried to make it more comprehensible and accomplish all of your requests.
The first figure was redone and the second figure has 7 quadrans and now are explained in the legend. We tried the explain the particular arrhythmic burden of the second patient, and to add more information about the mitral valve function in our patients.
We hope that the manuscript is in better shape now.

Round 2
Reviewer 2 Report
The paper has improved, but several english mispelling are still present: see as example "probant", "moderately regurgitation", "after de".
In material and methods some updated references of echocardiographic guidelines for standardized measurements should be reported.
In Figure 2 the echocardiographic parasternal long axis view can not be called "5 chambers".
Extensive english language editing required
Author Response
Please see the attachament
